# Mapping Pu'er tea plantations from GF-1 images using Object-Oriented Image Analysis (OOIA) and Support Vector Machine (SVM)

**Lei Liang[1,2,3], Jinliang Wang[1,2,3]\*, Fei Deng[1,2,3], Deyang Kong[1,2,3]**

**1** Faculty of Geography, Yunnan Normal University, Kunming, China, **2** Key Laboratory of Resources and Environmental Remote Sensing for Universities in Yunnan, Kunming, China, **3** Remote Sensing Research Laboratory, Center for Geospatial Information Engineering and Technology of Yunnan Province, Kunming, China

\* jlwang@ynnu.edu.cn

**Data Availability Statement:** All relevant data are within the manuscript and its Supporting information files.

## Abstract

Tea is the most popular drink worldwide, and China is the largest producer of tea. Therefore, tea is an important commercial crop in China, playing a significant role in domestic and foreign markets. It is necessary to make accurate and timely maps of the distribution of tea plantation areas for plantation management and decision making. In the present study, we propose a novel mapping method to map tea plantation. The town of Menghai in the Xishuangbanna Dai Autonomous Prefecture, Yunnan Province, China, was chosen as the study area, andgg GF-1 remotely sensed data from 2014–2017 were chosen as the data source. Image texture, spectral and geometrical features were integrated, while feature space was built by SEparability and THresholds algorithms (SEaTH) with decorrelation. Object-Oriented Image Analysis (OOIA) with a Support Vector Machine (SVM) algorithm was utilized to map tea plantation areas. The overall accuracy and Kappa coefficient ofh the proposed method were 93.14% and 0.81, respectively, 3.61% and 0.05, 6.99% and 0.14, 6.44% and 0.16 better than the results of CART method, Maximum likelihood method and CNN based method. The tea plantation area increased by 4,095.36 acre from 2014 to 2017, while the fastest-growing period is 2015 to 2016.

## Introduction

Tea is the most popular drink worldwide [1], and the plantation and production of tea play an important role in agricultural economies and country development. Furthermore, the excessive plantation of tea has many negative impacts on environments such as landslides and water and soil loss [2]; therefore, the continuous monitoring of tea-growing areas has significant importance.

Traditionally, related departments use visual interpretation methods to monitor tea distribution from remote sensing image, but these methods are inefficient, have a low precision, are expensive, and are challenged by the need for spatially continuous monitoring over large areas.

**Funding:** This work was supported by Multi-government International Science and Technology Innovation Cooperation Key Project of National Key Research and Development Program of China, grant number 2018YFE0184300; the National Natural Science Foundation of P.R. China, grant number 41561048; the Program for Innovative Research Team (in Science and Technology) in the University of Yunnan Province, IRTSTYN; and the Undergraduate Research Innovation Foundation of Yunnan Normal University, ky2017-072.

**Competing interests:** The authors have declared that no competing interests exist.

Remote sensing technology can obtain spatially continuous information on earth surfaces [3], which has plenty of applications in tea spatial distribution extraction and monitoring, because this technology has wide data coverage, fast data-update speeds, and low costs.

However, different tea growing areas have spectral differences due to canopy structure and traits, plantation size, crop health and growing period, enlarging the within-class distance of tea in spectral features. The spectral reflectance of tea is close to that of other vegetation, such as fruit trees and bushes, narrowing the class distance. Therefore, the spectral mixing phenomenon makes it extremely difficult to extract tea plantation distributions [4]. For example, He [5] used an artificial neural network (ANN) to classify vegetation using Landsat Thematic Mapper (TM) images found that the extraction of bushes and tea had a relatively low accuracy, but the extraction of other vegetation achieved a high accuracy. One effective method of plantation distribution extraction is to use spatial information based on texture features. For instance, Xu [3] built a decision tree based on the texture features of anisotropic strength to extract the tea plantation distribution in mountain areas, using ZY-3 satellite data. Dihkan [2] extracted the Gabor texture feature from multispectral airborne digital image to train a support vector machine classifier (SVM) and extract tea plantations distribution in black sea areas, achieving an overall accuracy above 90%. Fatemeh [6] used supervized neural network to extract and detect tea land losses based on Landsat 5–8 images. Another method is using time series analysis to extract temporal feature based on regular annual growth cycle of tea. For example, Li [7] used high-spatiotemporal-resolution VENµS images to extract temporal and spectral features, and utilized decision tree to mapping tea plantation. Zhu [8] integrates multi-temporal features to train a random forest classifier based on middle spital resolution satellite remote sensing data Sentinel-2. Wang [9] used multi-temporal landsat-8 imageries to extract multi-sseasonal feature and used random forest method classifier to map the tea plantaion. However, high-temporal-resolution means either low spatial resolution such as MODIS or small coverage such as VENµS, which not able to map extensive area. Besides, there are some method using Unmanned Aerial Vehicle (UAV) remote sensing data to map tea plantation. Zhang [10] used spectral and texture feature extracted from UAV image data combined with digital surface model (DSM) generated by UAV point clouds data to identify tea plantation. However, UAV data required field work and can not be used in large area. Because of data limitations, most previous studies based on mid/low-resolution satellite remote sensing data were unable to effectively extract texture features in tea-growing areas, while few high spatial resolution-based studies used aerial remote sensing data, with high costs and a relatively small coverage.

High spatial resolution satellite remote sensing data contain sufficient targets details, advancing the effect of the inside texture and structure features of targets. However, too exhaustive details may cause salt-and-pepper noise in information extraction and image classification. OOIA solves this problem effectively. For instance, Mallinis [11] used OOIA to extract the distribution of vegetation based on Quickbird satellite remote sensing data, which had a spatial resolution of 0.61 m and improved the classification accuracy. Chuang [12] used OOIA and machine learning to map tea plantation based on Worldview-2 remote sensing data. Tang [13] integrated OOIA with convolutional neural network to mapping tea plantation, achieving comparable results and can be used with limited training samples. However, Quickbird and Worldview-2 data are not free and public- available, and convolutional neural network sufered overfitting in small dataset.

Menghai town in the Xishuangbanna Dai Autonomous Prefecture, Yunnan Province, was chosen as the study area, and GF-1 remotely sensed data from 2014 to 2017 were chosen as the data source. In this study, OOIA and texture features were combined, while features selected by SEaTH, the CART algorithm and SVM algorithm were compared to find the best way to

extract the tea plantation area and analyze the plantation distribution and changes. In that case, the change trends of tea plantations can be analyzed to assist relevant departments in decision making.

## Study area and data source

### Study area

Menhai county is in the southwestern part of Yunnan province, the southwest corner of China. The longitude of this county is between 99˚56′E~100˚41′E, and the latitude is between 21˚28′N~22˚28′N (Fig 1). The study area has a tropical monsoon climate, with 1136–1513 ml precipitation per year. The annual average temperature is approximately 18.9˚C–22.6˚C, which is extremely suitable for tea growth, making this area one of the original plantations of "Pu Er tea". As of 2017, the overall tea plantation area was approximately 710 thousand acres, making the tea industry the largest and most profitable industry in Menhai County. Menhai town is in eastern Menhai County, which is the political, economy, and cultural center of Menhai County. The area of this town is $365.38km^2$, of which $45.3km^2$, is tea plantation area, which shows the tea industry plays a serious role in the local economy.

### Data source and preprocessing

The PMS(Panchromatic and multispectral sensors) data products from 2014 to 2017, which were obtained from the GF-1 satellite data, were used in the present study (Table 1). The pre-processes were supported by ENVI 5.3 software and consisting of seven steps. Primarily, radiometric calibration, FLAASH atmosphere correction, orthorectification, image pan sharping and image mosaicking were completed. Then, 2017 data were selected as the base image to register the other three images by image-to-image registering [14–17]. The RMS of the image registering result was controlled below 1 pixel. Finally, vector data of Menhai town were used to mask the image to obtain four scene images of Menhai town.

## Methods

### Research workflow

A six-steps method was used to map the tea plantations. First, image data were preprocessed by orthorectification, radiation calibration, atmospheric correction, seamless mosaic, image

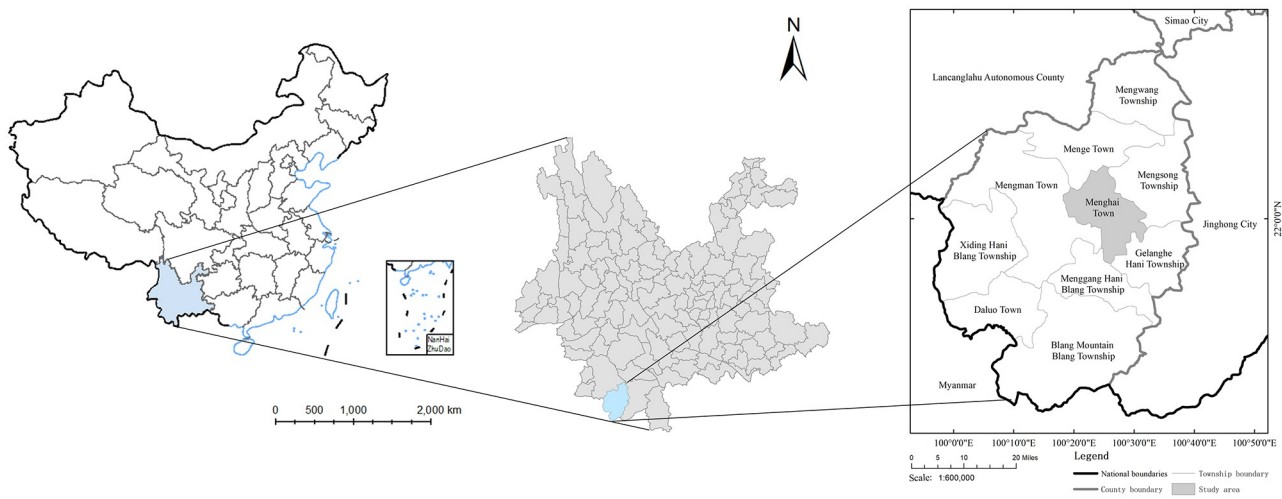

**Fig 1. Study area.**

**Table 1. GF-1 satellite PMS data parameters.**

| Parameters | 2 m pan/8 m multiresolution | |
|---|---|---|
| Spectral range | Pan | 0.45–0.90 μm |
| | Multispectral | 0.45–0.52 μm |
| | | 0.52–0.59 μm |
| | | 0.63–0.69 μm |
| | | 0.77–0.89 μm |
| Spatial resolution | Pan | 2 m |
| | Multispectral | 8 m |
| Swath | 60 km (2 camera) | |
| Temporal (sway) | 4 day | |
| Temporal (not sway) | 41day | |

registration and image mask. In the second step, we use threshold method based on normalized difference vegetation index (NDVI) to extract vegetation area. First, the NDVI was calculated from the preprocessed image. Second, OTSU [18] algorithm was used to select best NDVI threshold to separate non-vegetation area from vegetation area. The third step were multiresolution segmantation including selection of optimal segmentation and multiresolution segmentation. The fourth step was building feature space by the selection of best features using SEaTH. The fifth step was a comparison between the performing of SVM method, CART method, Maximum likelihood (ML) method and CNN based method to determine the best method. Finally, the best-performing method was utilized to extract tea distribution information from the rest of the image and analyze the changing situation (Fig 2).

## Extraction of vegetation

The NDVI is a well-performing index to indicate vegetation conditions on the surface. This index is obtained by the spectral reflectance of available light and near infrared between 400 and 860 nm because vegetation has special reflectance and absorption characteristics in these bands. The NDVI is calculated as follows:

$$NDVI = \frac{NIR - R}{NIR + R} \tag{1}$$

where NIR is the spectral reflectance of near infrared channel near 860μm, while R is the spectral reflectance of red channel near 660μm.

Because vegetation areas contain tea plantation areas, vegetation areas can be easily distinguished from non-vegetation areas by the NDVI threshold. In that case, the selection of the best-performing threshold is the most crucial stage.

The OTSU algorithm was proposed by Otsu [14], a Japanese researcher, in 1979, for image binarization, which was widely used to select the best threshold for grayscale segmentation in digital image processing. In this algorithm, the image is split into two classes, frontward and backward, by selecting the histogram value for segmentation that makes the variance highest between the two classes. In other words, this algorithm calculates the lowest point between two classifications in the histogram.

NDVI was calculated from preprocessed image data from 2014 to 2017 to select the optimized threshold (Table 2). Fig 3 shows the histogram of NDVI in 2017. The histogram can be considered having two peak forms, which makes the OTSU algorithm suitable. Fig 4 shows the results of the vegetation extraction for 2017, which achieved a quite accurate performance.

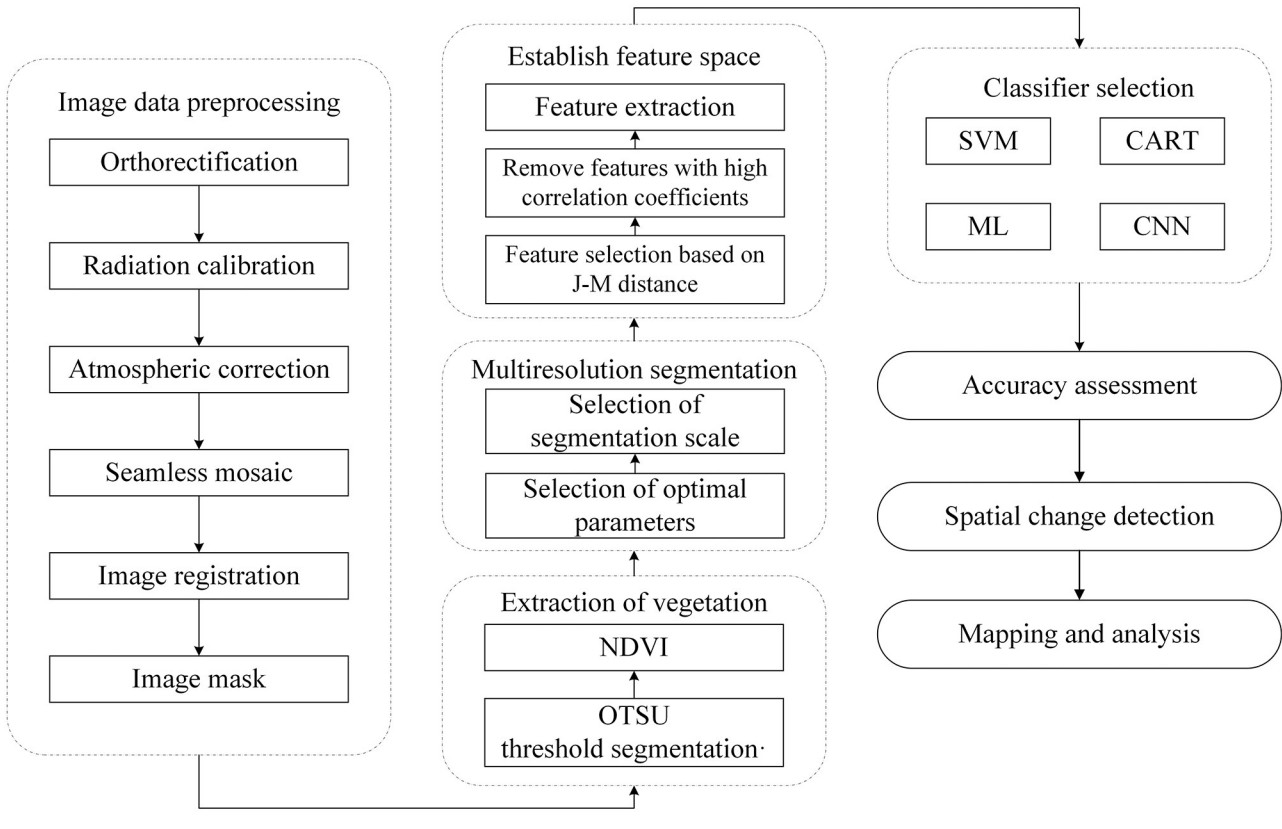

**Fig 2. Research workflow.**

## Selection of optimal parameters

To take advantage of the high-resolution feature and neighbor relationship of GF-1 satellite data and prevent noise, a multiresolution segmentation method, which combines neighbor pixels based on spectrum and shape to create image objects, was utilized [19]. In this stage, the selection of segmentation scale (SS), weight of spectrum (WS1), weight of shape (WS2), weight of smoothness (WS3), and weight of compactness (WC) controlled the performance of segmentation.

The estimation of scale parameters (ESP) is an optimal parameter selection method based on local variance (LV) and the rate of change (ROC) of image homogeneity [20]. LV is the average of the neighborhood standard deviation in the entire image. Along with the increase in SS, the inner homogeneity of the image object increased, and heterogeneity with other image objects decreased, causing the LV to rise and a slower change rate. When the optimal scale was achieved, the image objects had the highest similarity with earth surface features in reality, LV peaked, and ROC increased gradually. After visual assessment, the size and shape of the results were the most similar to the tea plantation area.

**Table 2. Optimal thresholds in 2014–2017.**

| year | 2014 | 2015 | 2016 | 2017 |
|---|---|---|---|---|
| threshold | 0.44 | 0.42 | 0.53 | 0.43 |

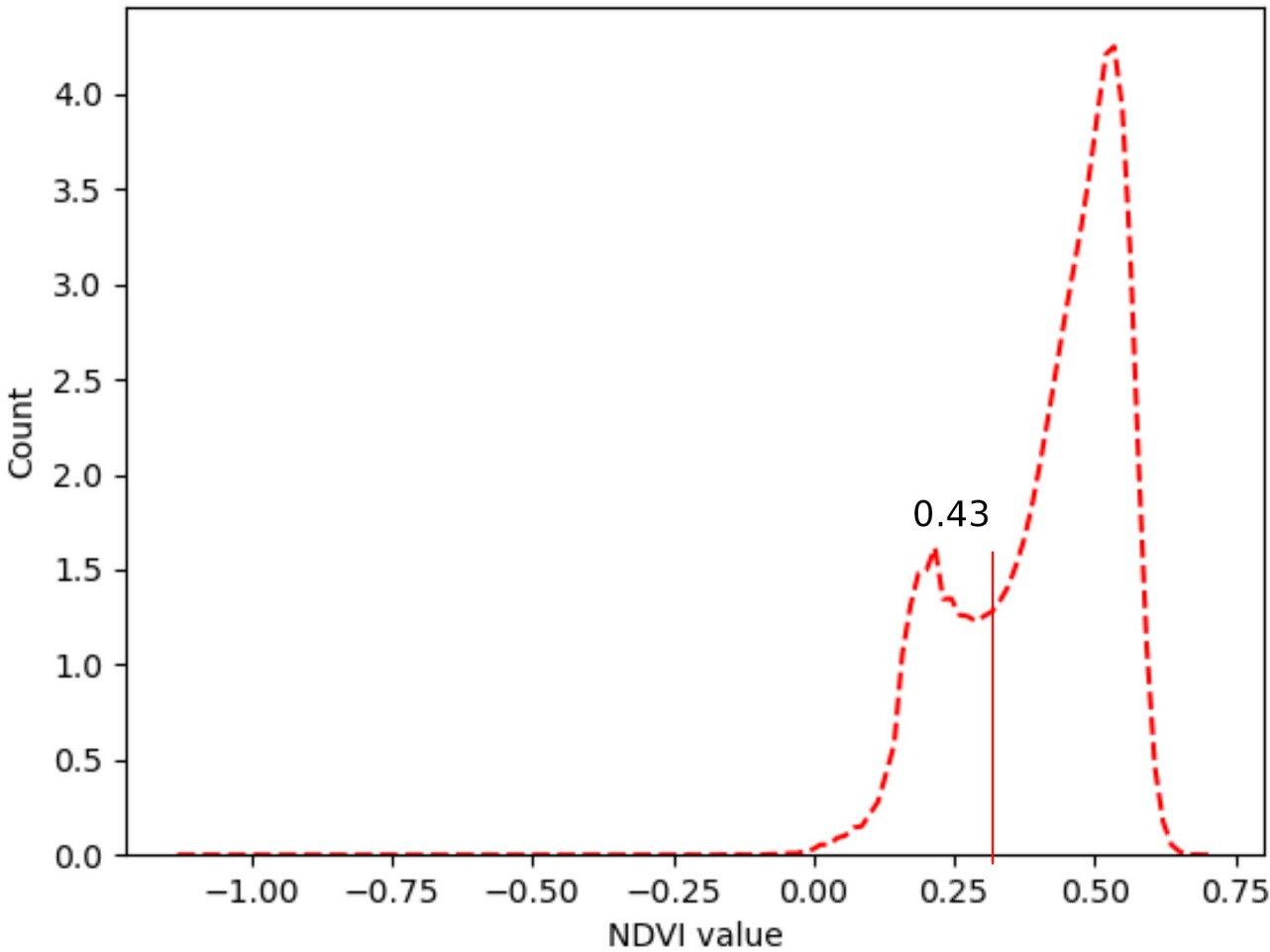

**Fig 3. NDVI histogram in 2017.**

In addition to the SS, the WS1 and WS2 control the segmentation results. WS2 also considers WS3 and WC. WS3 and WC are used to control the smoothness of the image object border and separation.

Based on the idea that the control variable is optimal, all SS and other parameters were traversed to integrate ESP and visual assessment to select optimal parameters. The results indicate that the optimal SS was 66, the optimal WS was 0.5, and the optimal WC was 0.7 (Fig 5).

When a larger SS than the optimal SS was used (Fig 6), e.g., 206, the area was too large, and details were missing, which can cause low accuracy in the classification step. When the SS was overlarge, the ROC of LV performing violent fluctuated, which is not the optimal parameter.

When a lower WC was used (Figs 7 and 8), the shape of the image object was too regular, and the border was too smooth, which was not the same as reality (Fig 7 (b)) and had a negative impact on segmentation. The optimal parameters resulted in the most complete extraction of tea plantation information, with distinct borders.

## Feature space establishment

The eCognition software offers hundreds of features, including spectrum, shape, texture, context, and spatial relationship features. The multidimensional feature space can solve the

## Vegetation Extraction Result in 2017

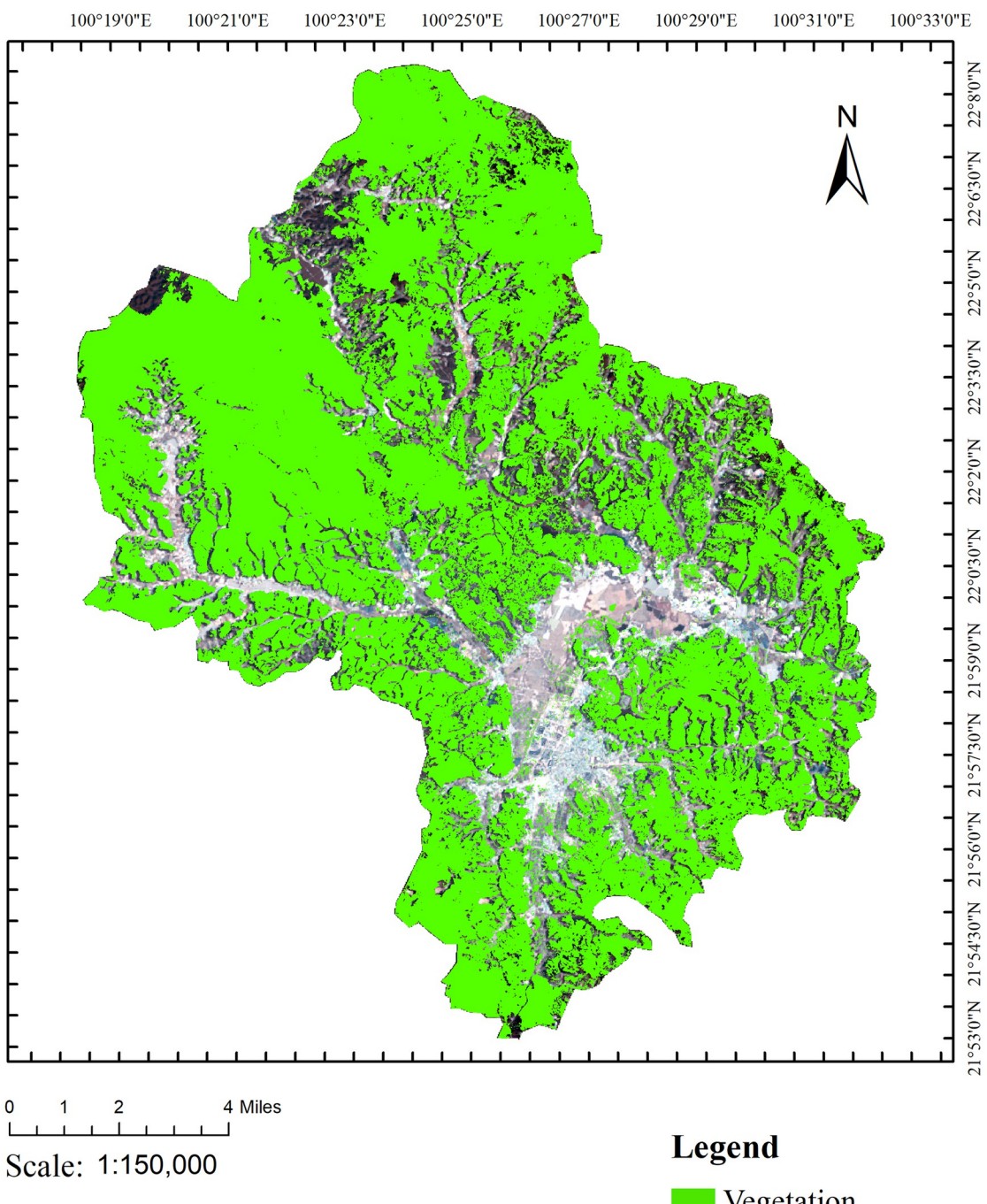

**Fig 4. 2017 Vegetation extraction results.** The base map of the LandSat8 OLI images with the following composition:R (4),G (3) and B (2). The LandSat8 OLI images were downloaded from USGS National Map Viewer.

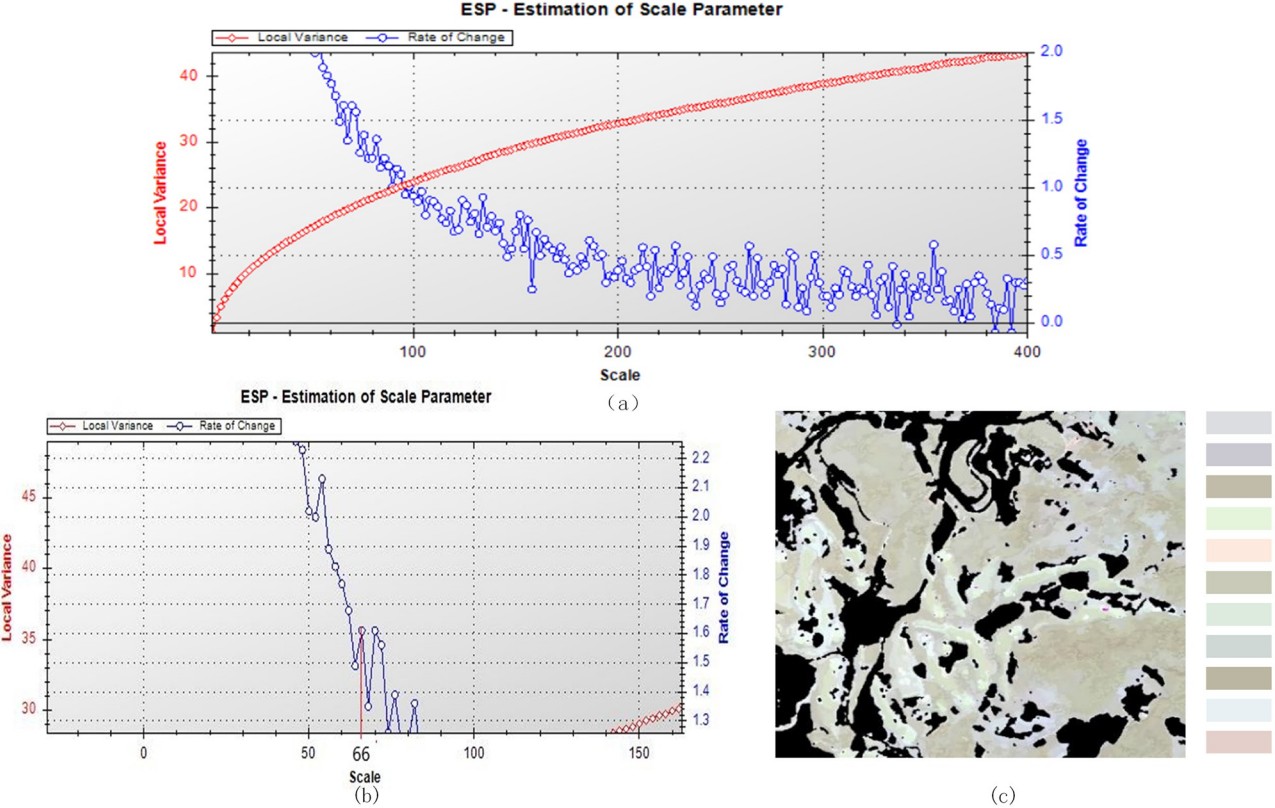

**Fig 5.** Multiresolution segmentation (SS = 66, WS_1 = 0.5, WC = 0.7): (a) variance curve, (b) optimal segmentation scale and (c) image object.

phenomenon caused by one-dimensional spectrum extraction. However, too many features can cause dimensional disasters, which damage calculation efficiency and accuracy [21]. Therefore, high efficiency and accuracy are crucial factors for the best selection from abundant features that have high discrimination and low redundancy.

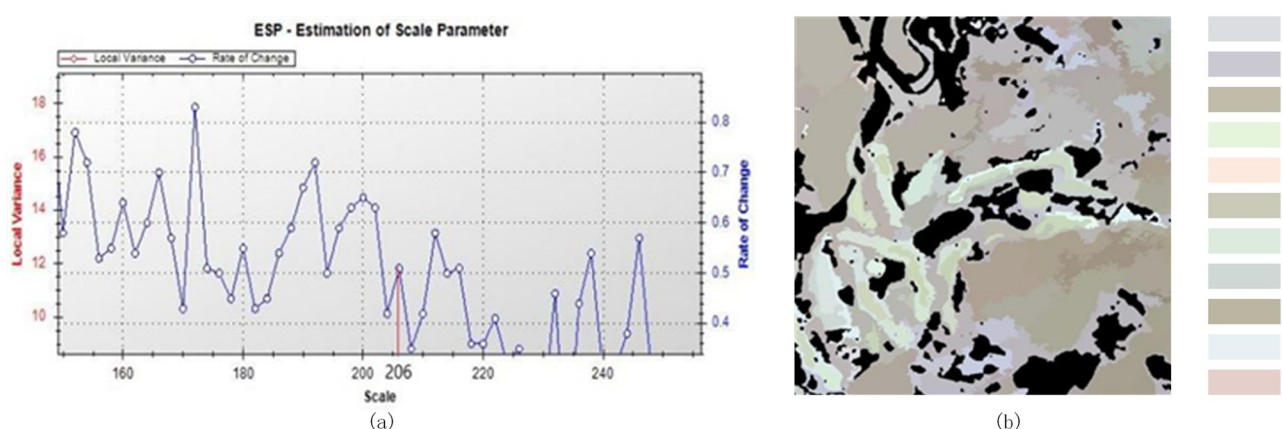

**Fig 6.** Multiresolution segmentation (SS = 206, WS_1 = 0.5, WC = 0.7): (a) variance curve and (b) image object.

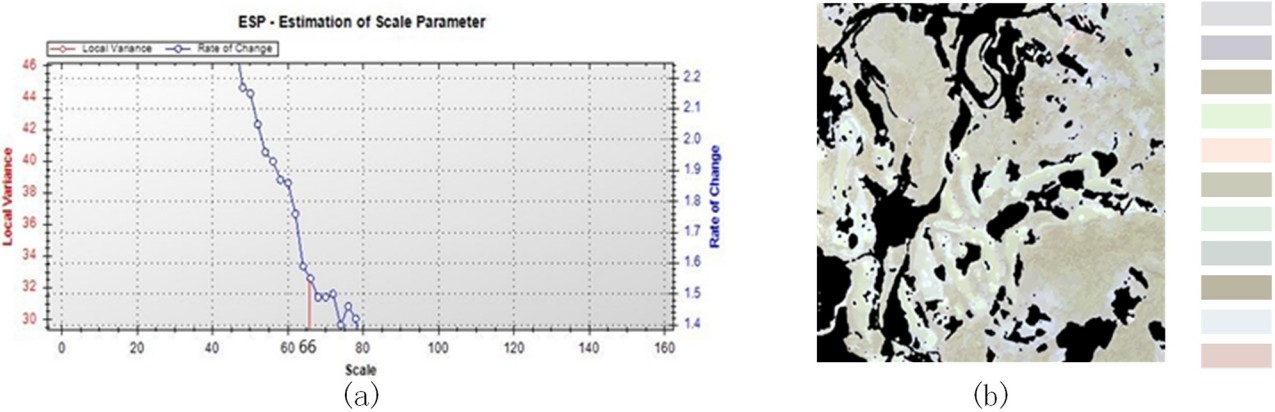

**Fig 7.** Multiresolution segmentation (SS = 66, WS_1 = 0.5, WC = 0.1): (a) variance curve and (b) image object.

The SEaTH algorithm is a feature selection method based on Jeffries Matusita(J-M) distance [21], which is calculated by the statistic index of two classes, such as the mean and variance, and used to measure discrimination in one feature dimensionality. A higher J-M distance value indicates better discrimination from the two class sample in one feature dimensionality. The J-M distance equation is:

$$J = 2(1 - e^{-B}) \tag{2}$$

$$B = \frac{1}{8(m_1 - m_2)^2} * \frac{2}{\sigma_1^2 + \sigma_2^2} + \frac{1}{2} ln \frac{\sigma_1^2 + \sigma_2^2}{2\sigma_1 \sigma_2} \tag{3}$$

where $m_i$ and $\sigma_i^2$ (i = 1, 2) refer to the mean and variance of a two-classification sample.

The range of the J-M distance is (0–2); 0 means completely mixed, while 2 means completely separated. Because the J-M distance is unable to completely reach 2, the best result is associated with the highest J-M distance.

Since multi-class classification often achieves better than binary classification. We split non-tea class by forest and cropland, and calculate J-M distance between tea and both of them.

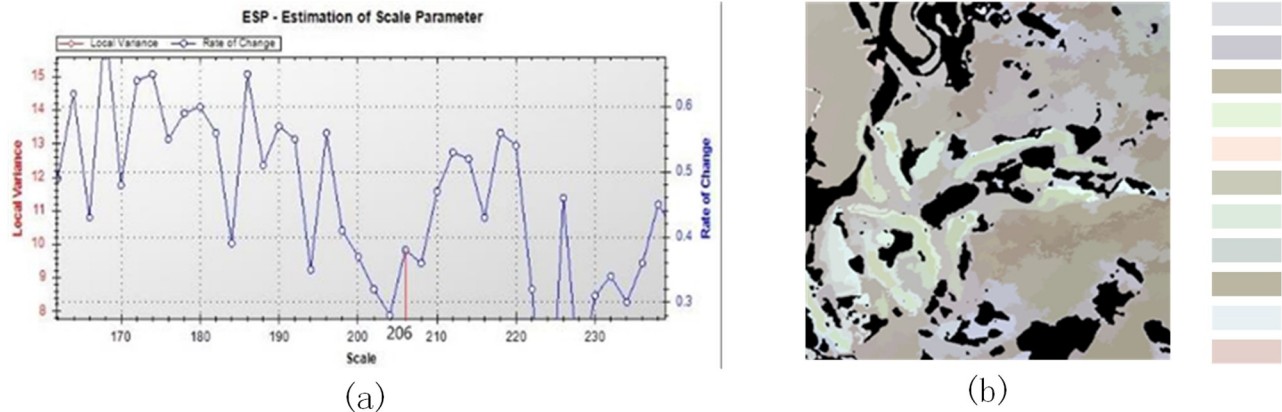

**Fig 8.** Multiresolution segmentation (SS = 206, WS_1 = 0.5, WC = 0.1): (a) variance curve and (b) image object.

**Table 3. J-M distances of part features (Homogeneity(Homo)).**

| | J-M distance (tea and corpland) | J-M distance (tea and forest) |
|---|---|---|
| Red band mean | 0.75 | 1.05 |
| GLCM Homo all dir | 0.44 | 0.84 |
| GLCM Homo 135 | 0.35 | 0.68 |
| Average area | 0.20 | 0.72 |
| Std of area | 0.53 | 0.68 |
| GLCM Homo 45 | 0.48 | 0.59 |
| Max. diff. | 0.20 | 0.75 |

Table 3 shows the J-M distance results calculated from different features, with results sorted by value.

Most features with high J-M distances are different operators based on the Gray-level cooccurrence matrix (GLCM) and Gray-level difference vector (GLDV). These features have strong discrimination but high correlation, which causes high redundancy and low accuracy. Thus, features with strong correlation coefficients should be removed. Pearson's correlation coefficient was utilized to calculate the correlation coefficients, as shown in Eqs 4 and 5:

$$r_{ij} = \frac{\sum_{l=1}^{K} \left( x_i^l - \overline{x_i} \right) \left( x_j^l - \overline{x_j} \right)}{\sqrt{\sum_{l=1}^{K} \left( x_i^l - \overline{x_i} \right)^2 \sum_{l=1}^{K} \left( x_j^l - \overline{x_j} \right)^2}} \tag{4}$$

$$\overline{x_i} = \frac{1}{K} \sum_{l=1}^{K} x_i^l \tag{5}$$

where $x_i^l$ refers to the i-th feature of the l-th sample and $\overline{x_i}$ refers to the mean value of the i-th feature.

Features with high correlation coefficients and relatively low J-M distances were removed from the 20 features with the highest J-M distances. The remaining features established a feature set with ten features (Table 4).

**Table 4. Correlation coefficients and J-M distances of feature set (homo(homogeneity) dis(dissimilarity), neig(neighbors)).**

| | GLCM homo | Width | GLDV mean | Max. diff | GLCM dis | GLCM StdDev | Num of neigh | Std of area | Red band mean | GLCM ASM |
|---|---|---|---|---|---|---|---|---|---|---|
| J-M distance (tea and forest) | 0.85 | 0.04 | 0.21 | 0.76 | 0.21 | 0.09 | 0.01 | 0.68 | 1.06 | 0.16 |
| J-M distance (tea and corp) | 0.44 | 0.86 | 0.76 | 0.20 | 0.76 | 0.78 | 0.91 | 0.54 | 0.75 | 0.89 |
| GLCM homo | 1.00 | 0.50 | -0.71 | 0.03 | -0.71 | -0.64 | 0.34 | 0.65 | -0.01 | -0.42 |
| Width | 0.50 | 1.00 | -0.63 | 0.40 | -0.63 | -0.72 | 0.81 | 0.53 | -0.49 | -0.34 |
| GLDV mean | -0.71 | -0.63 | 1.00 | -0.43 | 1.00 | 0.84 | -0.55 | -0.49 | 0.35 | 0.72 |
| Max. diff | 0.03 | 0.40 | -0.43 | 1.00 | -0.43 | -0.59 | 0.52 | 0.00 | -0.74 | -0.30 |
| GLCM dis | -0.71 | -0.63 | 1.00 | -0.43 | 1.00 | 0.84 | -0.55 | -0.49 | 0.35 | 0.72 |
| GLCM StdDev | -0.64 | -0.72 | 0.84 | -0.59 | 0.84 | 1.00 | -0.67 | -0.50 | 0.53 | 0.53 |
| Num of neigh | 0.34 | 0.81 | -0.55 | 0.52 | -0.55 | -0.67 | 1.00 | 0.34 | -0.58 | -0.31 |
| Std of area | 0.65 | 0.53 | -0.49 | 0.00 | -0.49 | -0.50 | 0.34 | 1.00 | -0.04 | -0.23 |
| Red band mean | -0.01 | -0.49 | 0.35 | -0.74 | 0.35 | 0.53 | -0.58 | -0.04 | 1.00 | 0.19 |
| GLCM ASM | -0.42 | -0.34 | 0.72 | -0.30 | 0.72 | 0.53 | -0.31 | -0.23 | 0.19 | 1.00 |

In this feature set, the most contributory features were texture features such as GLCM Angular Second Moment (ASM), GLCM homogeneity (homo), GLCM dissimilarity (dis). The texture feature is reflected by the correlation between two pixels at a certain distance and angle. Those features were calculated inside image object, using all pixels inside the image object. The equations of used texture features are shown (Eqs 6–10).

$$GLCM_{Homogeneity} = \sum_{i,j=0}^{N-1} P_{i,j}/\left(1 + (i-j)^2\right) \tag{6}$$

$$GLCM_{ASM} = \sum_{i,j=0}^{N-1} P_{i,j}{}^2 \tag{7}$$

$$GLCM_{Dissimilarity} = \sum_{i,j=0}^{N-1} P_{i,j}|i-j| \tag{8}$$

$$GLCM_{StdDev} = \sqrt{\sum_{i,j=0}^{N-1} P_{i,j}\left(i, j - u_{i,j}\right)} \tag{9}$$

$$GLDV_{Mean} = \sum_{i,j=0}^{N-1} k(V_k) \tag{10}$$

Because tea has similar spectral characteristics as other vegetation, the use of texture features can highly improve the accuracy and results of tea area extraction. Besides, the geometry features also have strong discrimination. The width of an image object is calculated using the length-to-width ratio. The Stddev of Area Represented by Segments is based on the standard deviation of triangles created by Delaunay triangulation. The number of neighbors calculated by the number of the direct neighbors of an image object. Finally, spectral features such as red band mean and max difference were used to build feature set. The Table 4 shows the best separate achievements in texture features.

## Classification method and accuracy assessment

Maximum likelihood algorithm, OOIA-based CART decision tree algorithm, CNN-based algorithm [13] and SVM algorithm were compared using the feature set established by the above methods to extract tea from 2017 image data. In case of CNN-based algorithm, features was obtain by CNN and selected by Gini index, and classifer was built by random forest, following the method proposed by Tang [13]. In case of SVM algorithm, the Radial Basis Function (RBF) was used as kernel function in order to deal with linear inseparable case.

The samples were gathered by field research and visual interpretation. The number and construction of samples were shown in Table 5. We split them for training and testing. So, the confusion matrix was calculated based on 2108 non-tea samples and 303 tea samples obtained by field observation to assess accuracy [22–24]. The best algorithm was chosen by comparing the overall accuracy, kappa coefficients, producer accuracy, and user accuracy to achieve the best performance during the extraction of tea area from multi-temporal image data.

Table 5. Training and validation samples.

| | Tea | Not tea | |
|---|---|---|---|
| | | Forest | Corpland |
| Training | 990 | 2927 | 2003 |
| Testing | 303 | 1254 | 854 |

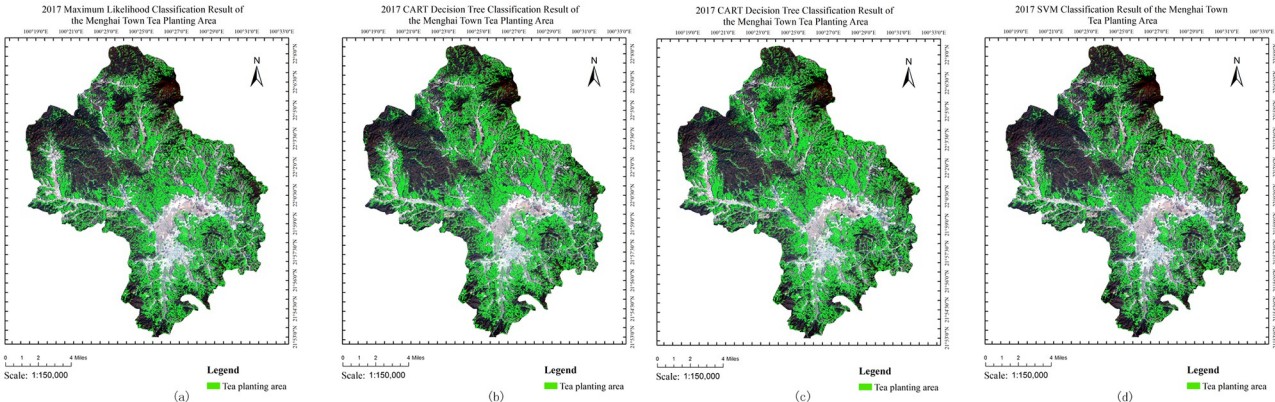

**Fig 9. Tea extraction result.** The base map of the LandSat8 OLI images with the following composition:R (4),G (3) and B (2). The LandSat8 OLI images were downloaded from USGS National Map Viewer.

## Results

### Extraction and result assessment

After the classification and accuracy assessment stage, tea plantation area (Figs 9 and 10) and accuracy (Table 6) were obtained. The CNN result achieve relatively low overall accuracy (86.7%) and lowest kappa coefficient (0.65) because of over-fitting problem in small sample case. The loss the training loss is converging to 0.01, while the validation loss is still not converging, causing low accuracy in validation set and even low accuracy in test set. Besides, because of the unbalanced number of less tea area and more other vegetation area, CNN model prefer to classify more other vegetation, causing low accuracy in tea plantation area extraction. The maximum likelihood results had the lowest user accuracy of tea (65.28%) because some other vegetation was mis-classified as tea. This misclassification was successfully

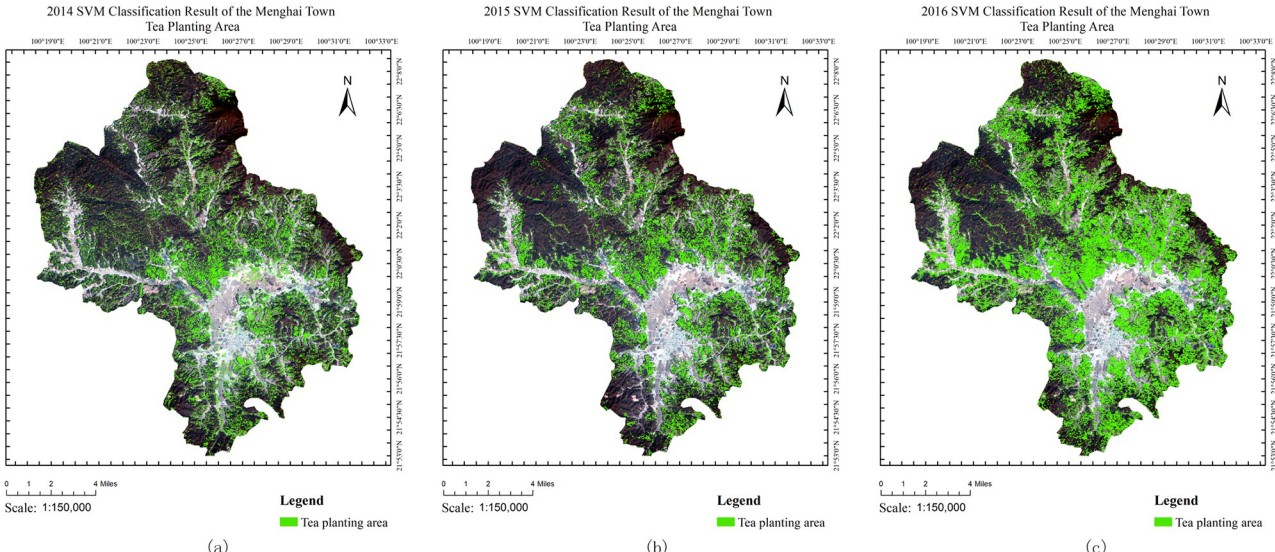

**Fig 10. Extraction results in (a)2014 (b)2015 and (c)2016.** The base map of the LandSat8 OLI images with the following composition:R (4),G (3) and B (2). The LandSat8 OLI images were downloaded from USGS National Map Viewer.

**Table 6. Accuracy assessment of 2017 classification result (OA(Overall accuracy), PA(Producer accuracy), UA(User accuracy), ML(Maximum likelihood)).**

|  | ML | CART | CNN | SVM |
|---|---|---|---|---|
| **OA** | 86.15% | 90.75% | 86.7% | 93.14% |
| **Kappa coefficient** | 0.67 | 0.76 | 0.65 | 0.81 |
| **PA (tea)** | 93.28% | 88.02% | 76.66% | 83.81% |
| **PA (other vegetation)** | 83.83% | 91.63% | 90.01% | 96.16% |
| **UA (tea)** | 65.28% | 77.43% | 71.43% | 87.7% |
| **UA (other vegetation)** | 97.45% | 95.91% | 92.2% | 94.81% |

**Table 7. Accuracy assessment of 2014–2016 SVM classification result (OA(Overall accuracy), PA(Producer Accuracy), UA(User Accuracy)).**

|  | 2014 | 2015 | 2016 |
|---|---|---|---|
| **OA** | 90.43% | 90.65% | 93.14% |
| **Kappa coefficient** | 0.71 | 0.75 | 0.81 |
| **PA (tea)** | 87.24% | 73.84% | 89% |
| **PA (other vegetation)** | 91.16% | 97.31% | 94.5% |
| **UA (tea)** | 69.23% | 91.6% | 84.17% |
| **UA (other vegetation)** | 96.6% | 90.37 | 96.31% |

minimized by the SVM algorithm, which improved the user accuracy of tea to 87.7%. However, the producer accuracy of tea of SVM (83.81%) results slightly lower then maximum likelihood (83.83%) and CART (88.02%) results. Since pine trees are widespread in the study area; which have similar the texture with tea, it is difficult to separate this vegetation from tea in optical remote sensing data. Thus, misclassification occurred between these two vegetation types, causing low accuracy, but the method could effectively separate tea from other vegetation, especially wood.

Then, proposed method was utilized to extract tea from three periods in 2014, 2015, and 2016, and the results contained tea-growing areas (Fig 10). To examine the accuracy of the results, the traditional accuracy assessment method was used to generate a confusion matrix (Tables 6 and 7). Some extracted tea areas were in the deep forest, but most tea was normally planted at the edge of a forest and in the neighborhood of the residential area during field research. However, the result accuracy was satisfactory for applications.

## Change detection

The ENVI software was utilized for change detection, while the area was calculated by Arc GIS software for statistical analysis. The spatial variation distribution (Fig 11) and area change situation were analyzed (Table 8, Figs 12 and 13). The overall pattern of spatial variation is expansion to the mid-west and east. The tea plantation area sharply increased between 2015 and 2016 and slowly increased between 2014 and 2015. The increasing rate was much slower between 2016 and 2017.

## Discussion

In this study, our primary objective was to analyze and propose a tea plantation mapping method and monitor the tea plantation area in Menghai Town from 2014 to 2017. The optimal segmentation parameters, such as segmentation scale (SS), weight of spectrum (WS1), weight

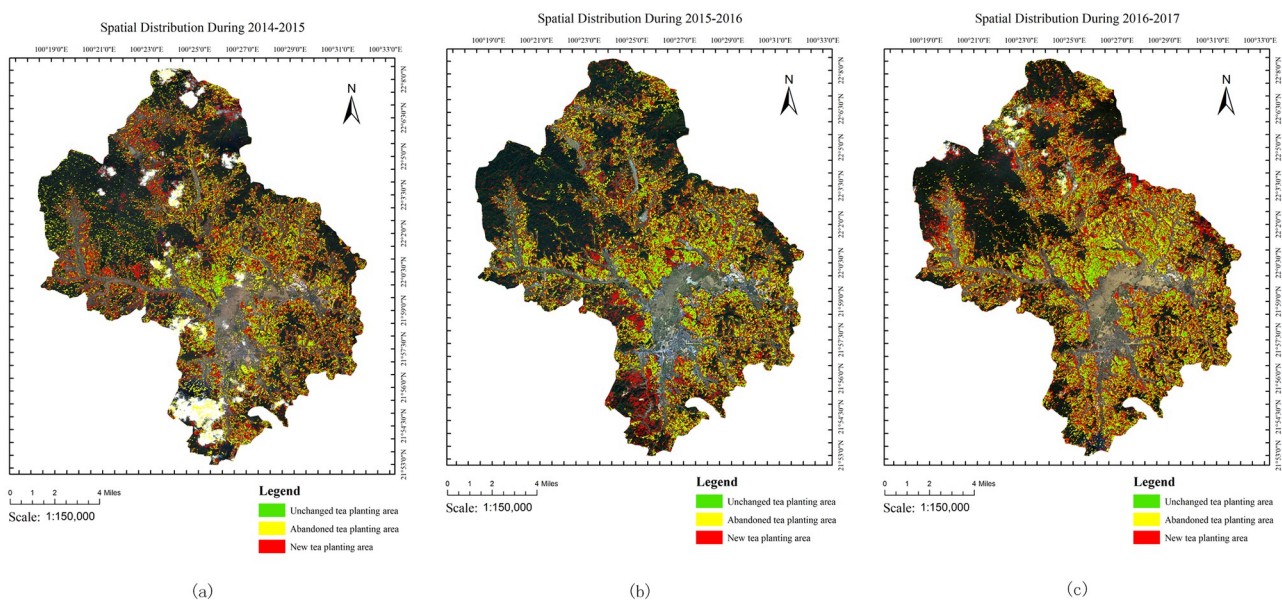

**Fig 11. Spatial distribution during (a)2014-2015 (b)2015-2016 and (c)2016-2017.** The base map of the LandSat8 OLI images with the following composition:R (4),G (3) and B (2). The LandSat8 OLI images were downloaded from USGS National Map Viewer.

**Table 8. Tea plantation area changes during 2014–2017.**

| Time | 2014 | 2015 | 2016 | 2017 |
|---|---|---|---|---|
| Area/acre | 13757.64 | 14705.58 | 17765.30 | 17853.00 |

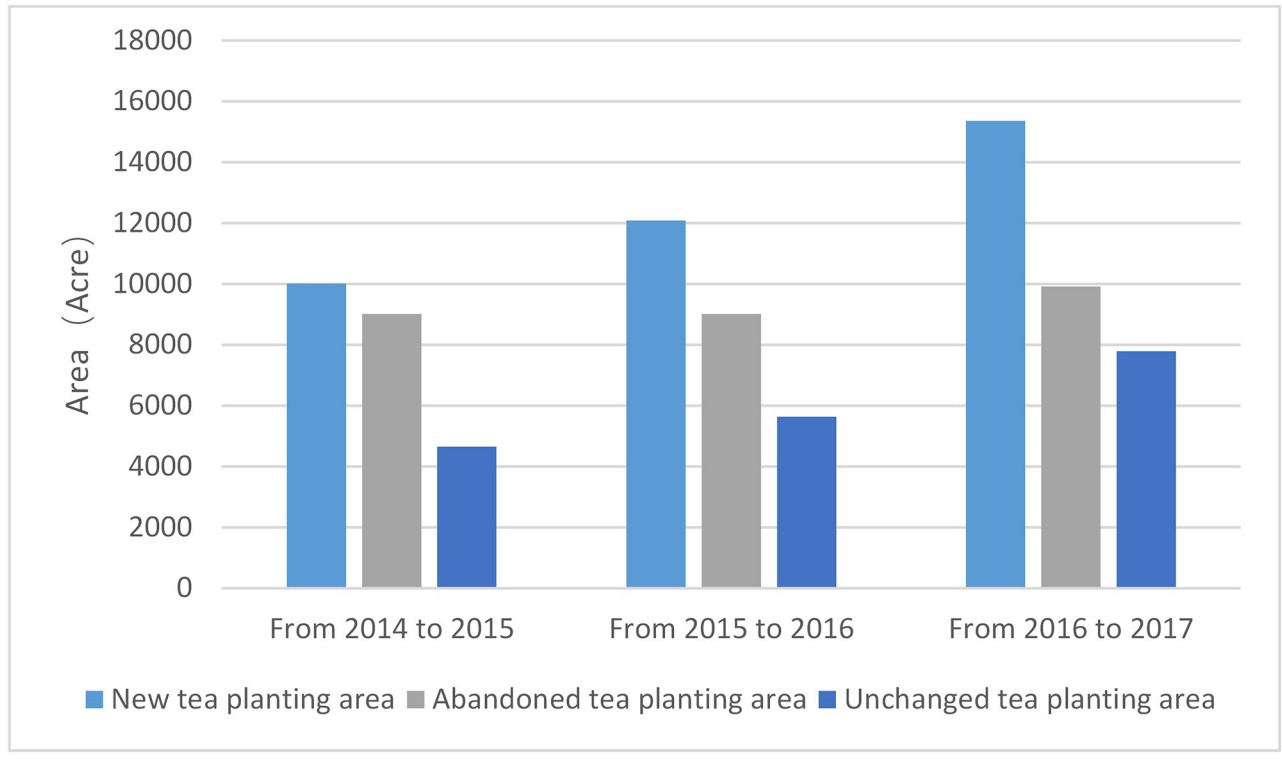

**Fig 12. Spatial variation from 2014 to 2017.**

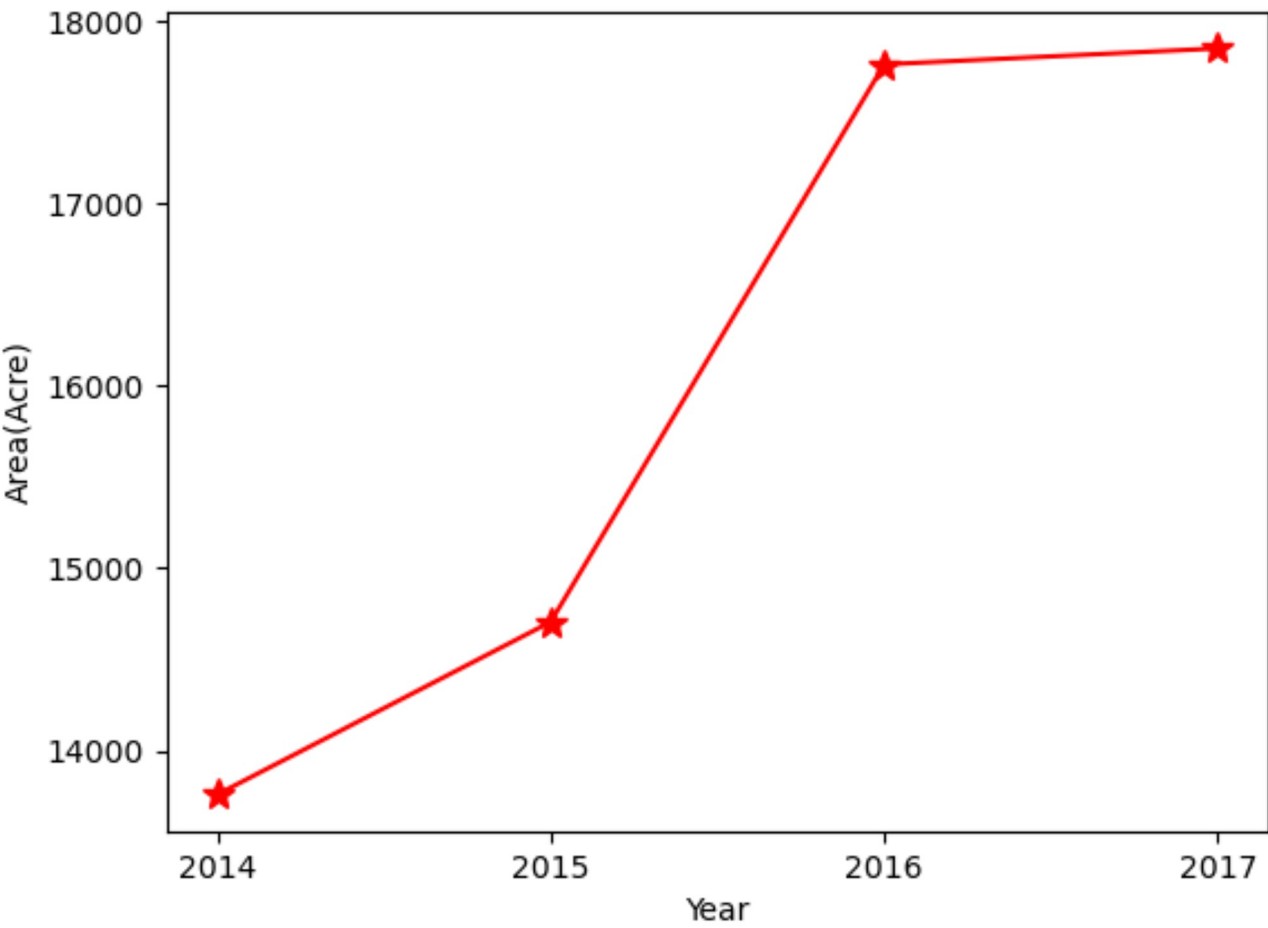

**Fig 13. Area change from 2014 to 2017.**

of shape (WS2), weight of smoothness (WS3), and weight of compactness (WC), were selected by calculate and analyze the local variance (LV) and the rate of change (ROC) of image homogeneity. The experiment results indicated that the optimal extraction of tea plantation are achieved when SS is set to 66, the WS is set to 0.5, and the WC is set to 0.7. The SS determines the size of the image object, which means higher SS causing larger image object and missing details. We found that for tea plantation mapping SS can not set too high because of the planting characteristics and vegetation form. WS1 is based on the standard deviation of the spectral colors, while WS2 is based on the deviation of a compact(or smooth) shape. WS1 determines to what degree spectrum influences the segmentation compared to shape, which against WS2. In Menghai town, most tea plantations have regular shape, so high WS2 can achieve better segmentation results. The value of WC gives it a relative weighting against WS3. The plantation form of tea in Menghai town is relatively compact, which make properly higher WC segment tea plantation better [25].

In order to build feature space for image objects to utilize the classification algorithm, SEaTH method is used to calculate the J-M distance of both spectrum and texture features between tea plantation and other vegetations. The experiment results showed that texture features, such as GLCM and GLDV, have the highest J-M distance. Since tea plantation in this area have more regular plantation form which significantly different with other vegetation,

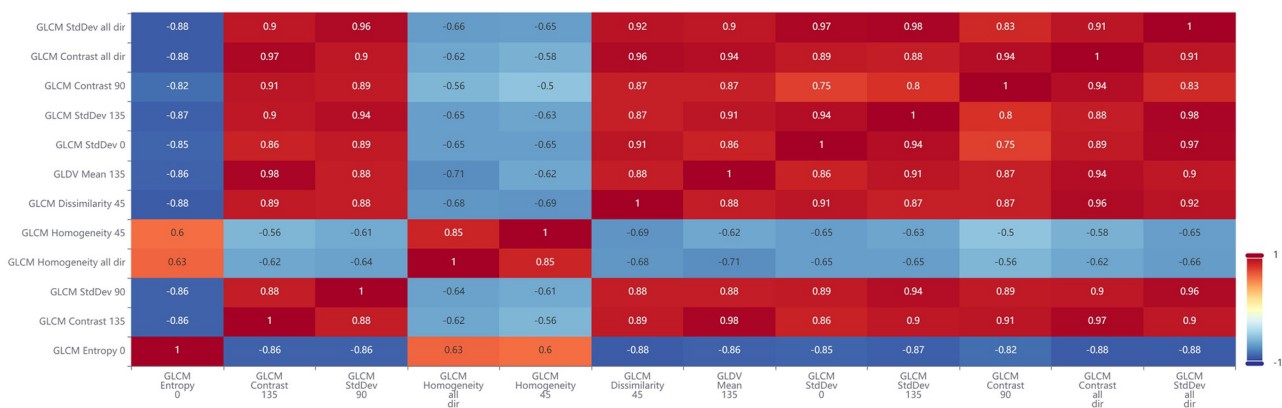

**Fig 14. Correlation coefficients of texture features.**

texture features are more suitable to separate tea from other vegetation. Besides, the mean value of the red band also have a relatively high J-M distance. Because vegetation coverage in tea plantation area normally lower than other vegetation in this area such as forest, the absorption effect of the red band in tea plantation area is relatively lower than other vegetation, causing high discrimination in the mean value of red band [26].

Then decorrelation method was used to remove the correlated features by calculating the correlation coefficients in order to build feature space without redundancy. The results indicated that although J-M distance of most texture features are high, the correlation coefficients are also high, which causes redundancy. Fig 14 shows the correlation coefficients of some texture features.

The accuracy of maximum likelihood algorithm, CRAT algorithm, CNN-based method [9] and SVM algorithm were evaluated at data in 2017 to find the most suitable algorithm. Besides, the performing of binary classification(tea and other vegetation) and multi-classification(tea, cropland and forest) was assessed. Table 9 shows the accuracy assessment results. SVM with multi-classification achieving 93.14% overall accuracy, 0.81 kappa coefficient and 87.7% user accuracy outperforming other methods. The experiment results also indicates that multi-classification method achieving better accuracy than binary classification method in all methods. Multi-classification CART improves 4.15% of overall accuracy and 9.97% of user accuracy on binary classification, which higher than SVM, because SVM only uses support vector to build classifier, which are more robust than CART in binary classification. Besides, multi-classification CNN slightly improves the accuracy of tea extraction on binary classification CNN, which are 0.01 of kappa coefficient and 4.16% of producer accuracy of tea.

**Table 9. Accuracy assessment of 2017 binary classification and multi-classification result (OA(Overall accuracy), PA(Producer accuracy), UA(User accuracy), ML (Maximum likelihood)).**

|  | ML(binary) | ML(multi) | CART (binary) | CART (multi) | CNN (binary) | CNN (multi) | SVM (binary) | SVM (multi) |
|---|---|---|---|---|---|---|---|---|
| **OA** | 82.5% | 86.15% | 86.6% | 90.75% | 86.9% | 86.7% | 91.94% | 93.14% |
| **Kappa** | 0.60 | 0.67 | 0.66 | 0.76 | 0.64 | 0.65 | 0.77 | 0.81 |
| **PA (tea)** | 93.35% | 93.28% | 86.60% | 88.02% | 72.5% | 76.66% | 79.87% | 83.81% |
| **PA (other vegetation)** | 78.97% | 83.83% | 86.40% | 91.63% | 91.6% | 90.01% | 95.87% | 96.16% |
| **UA (tea)** | 59.12% | 65.28% | 67.46% | 77.43% | 73.7% | 71.43% | 86.31% | 87.7% |
| **UA (other vegetation)** | 97.32% | 97.45% | 95.18% | 95.91% | 91.1% | 92.2% | 93.6% | 94.81% |

Then, the best performing algorithm was used to mapping tea plantation at 2014 to 2017. Accuracy assessment was done by confusion matrix and kappa coefficient (Table 6). All overall accuracy were above 90%, which proves the effectiveness of proposed method.

## Conclusion

Menghai Town of Xishuangbanna Dai Autonomous Prefecture, Yunnan Province, Chian,was chosen as the study area, and GF-1 remotely sensed data from 2014 to 2017 were chosen as the data source. Image texture, spectral and geometry features were integrated, while feature space was built by separability and thresholds algorithms (SEaTH) with decorrelation. Object-oriented image analysis (OOIA) with SVM algorithm was utilized to extract tea plantation areas. The salient findings of the study are as follows:

- OOIA can successfully suppress noise by object-based classification and improve the effect of extraction.

- The selection of segmentation scale and parameters is a critical factor in OOIA. LV and ROC can be used as criteria to evaluate parameters in order to generate meanful image objects. Scale parameter that is too small causes over segmentation, while a scale that is too large scale mixes different earth surface features in one image object.

- The feature space built by the SEaTH algorithm and decorrelation with texture features can effectively avoid spectral confusion.

- OOIA combined with the SVM algorithm and texture features solves the misclassification and missing-classification issues, yielding a Kappa coefficient up to 0.81 and an overall accuracy of 93.14%, higher than CART algorithm. Multi-classification method achieve better performance than binary classification in most algorithm.

- According to the extraction results, tea plantation areas are mainly distributed in the neighborhoods of residential areas with moderate slopes, because tea must be cultivated in easily drained areas and be convenient for farmers to reach. Based on variation analysis, we found that tea plantations expanded towards areas with sparse forests and moderate slopes.

## Expectations

The extraction method of tea plantations based on RS still needs to be discovered. Currently, methods still have flaws in accuracy and efficiency. In consideration of this, we will focus on the following aspect in feature research:

- ESP is utilized to select the segmentation scale, but other parameters still need to be selected based on a visual assessment. The parameter selection method can be improved by automatically selecting the weight of the spectrum and the weight of compactness based on LV to achieve the optimal segmentation result.

- In order to overcome texture and spectral confusion, different data and extraction methods can be used: consider the growing period of tea based on low revisit period RS data; multi-source data fusion such as hyperspectral and high-resolution RS data to analyze the spectral and texture features; integrate with LiDAR data to obtain canopy height and shape information to reduce spectrum fixing; use few sample learning method to overcome overfitting problem of deep learning and CNN.

## Supporting information

**S1 File. 2014–2017 Tea extraction result.**
(ZIP)

## Acknowledgments

The authors are grateful to the editor and anonymous referees for their valuable comments and helpful suggestions. We wish to thank professor Cheng Wang and Xiaohuang Xi for their advice.

## Author Contributions

**Conceptualization:** Jinliang Wang.

**Data curation:** Lei Liang, Fei Deng, Deyang Kong.

**Funding acquisition:** Jinliang Wang.

**Methodology:** Lei Liang.

**Project administration:** Jinliang Wang.

**Software:** Lei Liang, Fei Deng, Deyang Kong.

**Supervision:** Jinliang Wang.

**Visualization:** Lei Liang.

**Writing – original draft:** Lei Liang.

**Writing – review & editing:** Lei Liang, Jinliang Wang.

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
