## [Decision Letter · Decision Letter 0]

23 Jun 2021

PONE-D-21-11402

Mapping Pu'er tea plantations from GF-1 images using Object-oriented image analysis (OOIA) andclassification and regression tree (CART)

PLOS ONE

Dear Dr. Wang,

Thank you for submitting your manuscript to PLOS ONE. After careful consideration, we feel that it has merit but does not fully meet PLOS ONE’s publication criteria as it currently stands. Therefore, we invite you to submit a revised version of the manuscript that addresses the points raised during the review process.

One of the reviewers suggested reject whereas two suggested revision. Please, carefully address the publication requirements of the three reviewers in order to proceed with the eventual acceptance. 

We look forward to receiving your revised manuscript.

Kind regards,

Daniel Capella Zanotta

Academic Editor

PLOS ONE

Journal Requirements:

'This work was supported by Multi-government International Science and Technology Innovation Cooperation Key Project of National Key Research and Development Program of China，grant number 2018YFE0184300; the National Natural Science Foundation of P.R. China, grant number 41561048; the Program for Innovative Research Team (in Science and Technology) in the University of Yunnan Province, IRTSTYN; and the Undergraduate Research Innovation Foundation of Yunnan Normal University. The authors are grateful to the editor and anonymous referees for their valuable comments and helpful suggestions. We wish to thank professor Cheng Wang and Xiaohuang Xi for their advice.'

'The author(s) received no specific funding for this work.'

Additional Editor Comments (if provided):

Reviewers' comments:

Reviewer's Responses to Questions

**Comments to the Author**

1. Is the manuscript technically sound, and do the data support the conclusions?

Reviewer #1: Partly

Reviewer #2: Yes

Reviewer #3: Yes

2. Has the statistical analysis been performed appropriately and rigorously? 

Reviewer #1: No

Reviewer #2: Yes

Reviewer #3: Yes

3. Have the authors made all data underlying the findings in their manuscript fully available?

Reviewer #1: No

Reviewer #2: Yes

Reviewer #3: Yes

4. Is the manuscript presented in an intelligible fashion and written in standard English?

Reviewer #1: No

Reviewer #2: No

Reviewer #3: Yes

5. Review Comments to the Author

Reviewer #1: This paper use a kind of so called six-step method to extract the planting area of Pu'er tea from GF-1 images. The planting area were extracted by using the object-base CART with the texture, spectral and spatial features. There are some suggestions:

(1) The title of this paper need to be revised. "Mapping Pu'er tea plantations from GF-1 images using Object-oriented image analysis (OOIA) andclassification and regression tree (CART)". A space character shoud be inserted between the word "and" and "classification"!!!

(2) In the abstract, "The overall accu-racy and Kappa coefficient". Why the "-" is inserted into the word "accuracy"? There are many similar mistakes in the full text. Please revise these errors carefully！！

（3）In line 88, "of this town is 365.38,, of which 45.3 88 ,..." the redundant commas should be removed!!

(4) In line 108, "from vegetation area Third step was multiresolution segmentation", where is the full stop between “area” and "Third"??

(5)The language of the paper needs to be retouched！

（6）Why do you just compare the performance of maximum likelihood and CART? why not support verctor machine or the convolutional neural network (CNN) as you mentioned in the introduction section. CNN has been widely used in crop planting area extraction. What are the advantages of your method？

(7) The legends should be added in Figure 6-8 to illustrate the segmentation results.

Reviewer #2: Dear authors,

This is a good work. However, there are some deficiencies which have to be considered. It can improve the research paper.

- Throughout the paper, please carefully check and correct grammatical errors and typos.

- Line 120, some explanations about NDVI should be revised. Please see my comment in the attached file.

- Line 96, some of the technical terms are misspelled.

- The main drawback of the paper is the lack of appropriate discussion. Some presented reasons about why these results were achieved are not convincing. Discussion section provides a summary of work instead of discussion.

- Details of texture implementation such as window size used are not mentioned.

- Details of the accuracy assessment, such as the number of training samples and how they were gathered, are not given.

Please find the attached file. I included all of my comments in the text.

Regards.

Reviewer #3: Journal: PLOS ONE

Manuscript ID: PONE-D-21-11402

Mapping Pu'er tea plantations from GF-1 images using Object-oriented image analysis (OOIA) and classification and regression tree (CART)

The manuscript has good scientific results on the maps of the distribution of tea plantation areas to for plantation management and decision making. The present paper presents the satellite-based production proxies, such as GF-1 images using Object-oriented image analysis (OOIA) and classification and regression tree (CART) to extract tea plantation area during 2014-2017, in the study area of the town of Menghai in the Xishuangbanna Dai Autonomous Prefecture, Yunnan Province. The publication could be an excellent contribution to the journal. The paper can be considered for publication in the Journal of PLOS ONE after minor corrections.

Reviewer commands:

To improve the manuscript, please provide the following data and changes as follows:

1) The authors should read the manuscript and clarify the command clearly. The authors have to care about typos. There are careless mistakes in some places. Example: environ-ments, moni-toring, distribu-tion, accu-racy, …

2) The quality of Figure 1 is poor. Please improve it.

3) L105-114: The authors should better explain better the methodology to analyze the tea plantations.

4) The authors mentioned that all SS and other parameters were traversed to integrate ESP and visual assessment to select optimal parameters. Please explain the correlation between the SS, WS, and WC parameters? Add some references.

5) L306-312: The discussion section should be the focus on each of the parametric discussed as confidential factors. You can add some references. Remote Sensing. 2020, 12, 3869. https://doi.org/10.3390/rs12233869; Envi. Sci. Pull. Resea. 27(6):5873-5889. DOI:10.1007/s11356-019-07216-1

6) The quality of all the figures should be improved.

7) In the Conclusion section, the authors should be re-writing some sentences (L323-L353).

The reviewer commands the paper to be accepted after minor revision.

I congratulate the authors for the quality work.

6. PLOS authors have the option to publish the peer review history of their article (what does this mean?). If published, this will include your full peer review and any attached files.

Reviewer #1: No

Reviewer #2: No

Reviewer #3: **Yes: **Malak Henchiri

---

## [Author Response · Author response to Decision Letter 0]

11 Aug 2021

Thanks for the important comments and suggestions from reviewers and editor. Please find the attached file to check the responses.

---

## [Editor Report · Decision Letter 1]

31 Aug 2021

PONE-D-21-11402R1

Mapping Pu'er Tea Plantations from GF-1 Images Using Object-Oriented Image Analysis (OOIA) and Support Vector Machine (SVM)

PLOS ONE

Dear Dr. Wang,

Thank you for submitting your manuscript to PLOS ONE. After careful consideration, we feel that it has merit but does not fully meet PLOS ONE’s publication criteria as it currently stands. Therefore, we invite you to submit a revised version of the manuscript that addresses the points raised during the review process.

We look forward to receiving your revised manuscript.

Kind regards,

Daniel Capella Zanotta

Academic Editor

PLOS ONE
---

## [Author Response · Author response to Decision Letter 1]

4 Oct 2021

Dear Reviewer:

 Thank you for your comments concerning our manuscript entitled “Mapping Pu'er tea plantations from GF-1 images using OOIA and SVM” (ID: PONE-D-21-11402R1). Those comments are valuable and helpful for revising and improving our paper, as well as the important guiding significance to our researches. We have studied comments carefully and have made correction which we hope meet with approval. The responses to the reviewer’s comments can be found in attachment files.

---

## [Decision Letter · Decision Letter 2]

9 Nov 2021

PONE-D-21-11402R2Mapping Pu'er Tea Plantations from GF-1 Images Using Object-Oriented Image Analysis (OOIA) and Support Vector Machine (SVM)PLOS ONE

Dear Dr. Wang,

Thank you for submitting your manuscript to PLOS ONE. After careful consideration, we feel that it has merit but does not fully meet PLOS ONE’s publication criteria as it currently stands. Therefore, we invite you to submit a revised version of the manuscript that addresses the points raised during the review process.

We look forward to receiving your revised manuscript.

Kind regards,

Daniel Capella Zanotta

Academic Editor

PLOS ONE

Journal Requirements:

Additional Editor Comments (if provided):

Dear Authors,

The reviewers have once again analysed your paper and at this time they consider the work acceptable for publication with minor revisions. Indeed, two of them have already accepted the actual current content as it is.

Reviewers' comments:

Reviewer's Responses to Questions

**Comments to the Author**

1. If the authors have adequately addressed your comments raised in a previous round of review and you feel that this manuscript is now acceptable for publication, you may indicate that here to bypass the “Comments to the Author” section, enter your conflict of interest statement in the “Confidential to Editor” section, and submit your "Accept" recommendation.

Reviewer #1: All comments have been addressed

Reviewer #2: All comments have been addressed

Reviewer #3: All comments have been addressed

2. Is the manuscript technically sound, and do the data support the conclusions?

Reviewer #1: Yes

Reviewer #2: Yes

Reviewer #3: Yes

3. Has the statistical analysis been performed appropriately and rigorously? 

Reviewer #1: Yes

Reviewer #2: Yes

Reviewer #3: Yes

4. Have the authors made all data underlying the findings in their manuscript fully available?

Reviewer #1: Yes

Reviewer #2: Yes

Reviewer #3: Yes

5. Is the manuscript presented in an intelligible fashion and written in standard English?

Reviewer #1: Yes

Reviewer #2: Yes

Reviewer #3: Yes

6. Review Comments to the Author

Reviewer #1: (No Response)

Reviewer #2: Thanks to the authors for considering the reviewers' comments. The paper has been improved. All my comments have fully been addressed.

There are still some minor grammatical mistakes, for example line 75 of the revised version. Please see the attached file.

Please give the full name of the acronyms in line 40 of the revised version (CART and CNN).

Since the SVM method has newly been added to the work, please explain how you found the optimum values of its parameters (i.e. the penalty parameter(c) and the width of Gaussian function of RBF (gamma) ). Please see line 270 of the revised version.

Best regardes.

Reviewer #3: Manuscript Number: PONE-D-21-11402R2

Manuscript Title: Mapping Pu'er Tea Plantations from GF-1 Images Using Object-Oriented Image Analysis (OOIA) and Support Vector Machine (SVM).

I would like to thank the authors for their answers. the following manuscript has been improved and can be accepted after a minor revision.

1. The authors should revise carefully the format of the manuscript; the size of titles is too large; the authors should follow the guidelines for authors. Also they can add the numbers for the titles and sub-titles in the manuscript.

2. The authors should revise all the tables especially tables 4 and 9.

7. PLOS authors have the option to publish the peer review history of their article (what does this mean?). If published, this will include your full peer review and any attached files.

Reviewer #1: No

Reviewer #2: No

Reviewer #3: No

---

## [Author Response · Author response to Decision Letter 2]

22 Nov 2021

Dear Editors and Reviewer:

 Thank you for your comments concerning our manuscript entitled “Mapping Pu'er tea plantations from GF-1 images using OOIA and SVM” (ID: PONE-D-21-11402R1). Those comments are valuable and helpful for revising and improving our paper, as well as the important guiding significance to our researches. We have studied comments carefully and have made correction which we hope meet with approval. The responses to the reviewer’s comments can be found in attchements

---

## [Decision Letter · Decision Letter 3]

8 Dec 2021

PONE-D-21-11402R3Mapping Pu'er Tea Plantations from GF-1 Images Using Object-Oriented Image Analysis (OOIA) and Support Vector Machine (SVM)PLOS ONE

Dear Dr. %Wang%,

Thank you for submitting your manuscript to PLOS ONE. After careful consideration, we feel that it has merit but does not fully meet PLOS ONE’s publication criteria as it currently stands. Therefore, we invite you to submit a revised version of the manuscript that addresses the points raised during the review process.

Two reviewers have already accepted the work. However several issues from the third reviewer had not been addressed yet.

We look forward to receiving your revised manuscript.

Kind regards,

Daniel Capella Zanotta

Academic Editor

PLOS ONE

Journal Requirements:

Additional Editor Comments (if provided):

Before the authors resubmit this manuscript to this journal with minor issues addressed ,please carefully consider the requests of this reviwer. Otherwise, we cannot proceed with the publication if the authors do not satisfactorily answer the raised points.

Reviewers' comments:

Reviewer's Responses to Questions

**Comments to the Author**

1. If the authors have adequately addressed your comments raised in a previous round of review and you feel that this manuscript is now acceptable for publication, you may indicate that here to bypass the “Comments to the Author” section, enter your conflict of interest statement in the “Confidential to Editor” section, and submit your "Accept" recommendation.

Reviewer #1: All comments have been addressed

Reviewer #2: All comments have been addressed

Reviewer #3: (No Response)

2. Is the manuscript technically sound, and do the data support the conclusions?

Reviewer #1: Yes

Reviewer #2: Yes

Reviewer #3: No

3. Has the statistical analysis been performed appropriately and rigorously? 

Reviewer #1: Yes

Reviewer #2: Yes

Reviewer #3: No

4. Have the authors made all data underlying the findings in their manuscript fully available?

Reviewer #1: Yes

Reviewer #2: Yes

Reviewer #3: No

5. Is the manuscript presented in an intelligible fashion and written in standard English?

Reviewer #1: Yes

Reviewer #2: Yes

Reviewer #3: No

6. Review Comments to the Author

Reviewer #1: The comments from the reviewers were replied and the quality of the paper is improved significantly. Thus the paper is eligible.

Reviewer #2: Thanks for considering my comments. I have no more comment. In my opinion the paper is ready to be published.

Reviewer #3: Manuscript ID: PONE-D-21-11402R3

Title: Mapping Pu'er Tea Plantations from GF-1 Images Using Object-Oriented Image Analysis (OOIA) and Support Vector Machine (SVM)

PLOS ONE

I do not feel that this article is suitable for publishing in PLOS ONE Journal because the revision is so weak and the manuscript still has many gaps. Before the authors resubmit this manuscript to this journal or another journal please carefully consider my comments.

Recommendations for Authors:

1. In the abstract, the authors should add the manuscript's novelty and topicality.

2. The authors could rearrange the keywords alphabetically.

3. The authors should improve the introduction section, add recently published papers (after 2016), mention the gaps in previous studies and the innovations in this study and clarify the aims.

4. Weak methodology, please improve this part and clearly explain the methodology used in this study.

5. All the Figures are not clear, please plot your figure according to your results don’t take ready Figure from the output of program.

6. Change the Figure of the study area to a new figure that shows clearly the location of the study area.

7. Improve the methodology flowchart.

8. The authors should re-arrange all the Tables in the revised manuscript. Example like in Table 9 and 4 the authors could use abbreviation.

9. Please add more recently published papers in your revised manuscript. And keep the same format for all the references.

10. The authors should clearly discuss the results and compare them with previous work.

11. Future research statement should be stated in the conclusion section.

7. PLOS authors have the option to publish the peer review history of their article (what does this mean?). If published, this will include your full peer review and any attached files.

Reviewer #1: No

Reviewer #2: No

Reviewer #3: No

---

## [Author Response · Author response to Decision Letter 3]

2 Jan 2022

Response to Reviewer #3 comments:

Dear Reviewer:

 Thank you for your comments concerning our manuscript entitled “Mapping Pu'er tea plantations from GF-1 images using OOIA and SVM” (ID: PONE-D-21-11402R1). Those comments are valuable and helpful for revising and improving our paper, as well as the important guiding significance to our researches. We have studied comments carefully and have made correction which we hope meet with approval. The responses to the reviewer’s comments are as follows:

* 1. In the abstract, the authors should add the manuscript's novelty and topicality.

Response 1: Thanks for your suggestion. We have added the manuscript's novelty and topicality in abstract.

* 2. The authors could rearrange the keywords alphabetically.

Response 2: Thanks for you remind. We have rearranged the keywords.

* 3. The authors should improve the introduction section, add recently published papers (after 2016), mention the gaps in previous studies and the innovations in this study and clarify the aims.

Response 3: Thanks for your suggestion. We have added more related papers in introduction, for example, Wang B, Li J, Jin X, et al. Mapping Tea Plantations from Multi-seasonal Landsat-8 OLI Imageries Using a Random Forest Classifier. J Indian Soc Remote Sens. 2019; 47, 1315–1329; Fatemeh R, Mahdi K, Remote sensing-based detection of tea land losses: The case of Lahijan, Iran. Remote Sensing Applications: Society and Environment. 2021; 23: 568; Zhang Q, Wan B, Cao Z, Zhang Q. Exploring the Potential of Unmanned Aerial Vehicle (UAV) Remote Sensing for Mapping Plucking Area of Tea Plantations. Forests. 2021; 12(9):1214. We have added the gaps in previous studies and the innovations in this study. 

* 4. Weak methodology, please improve this part and clearly explain the methodology used in this study.

Response 4: Thanks for your suggestion. We have improved the Research workflow part to better explain the methodology.

* 5. All the Figures are not clear, please plot your figure according to your results don’t take ready Figure from the output of program.

Response 5: Thanks for your suggestion. We have improved the clarity of all figures and re-plot the Fig 3 and Fig 12-13.

* 6. Change the Figure of the study area to a new figure that shows clearly the location of the study area.

Response 6: Thanks for your suggestion. We have re-plot the study area figure Fig 1.

* 7. Improve the methodology flowchart.

Response 7: Thanks for your suggestion. We have improved the methodology flowchart.

* 8. The authors should re-arrange all the Tables in the revised manuscript. Example like in Table 9 and 4 the authors could use abbreviation.

Response 8: Thanks for your suggestion. We have re-arranged tables in the manuscript.

* 9. Please add more recently published papers in your revised manuscript. And keep the same format for all the references.

Response 9: Thanks for your suggestion. We have added more recently published papers and the references.

* 10. The authors should clearly discuss the results and compare them with previous work.

Response 10: Thanks for you suggestion. We have improved the discussion of the results, and compared them with maximum likelihood, CART (Xu, 2016), CNN (Zixia, 2020).

* 11. Future research statement should be stated in the conclusion section.

Response 11: Thanks for your suggestion. We have added the future research statement in the expectation section.

---

## [Decision Letter · Decision Letter 4]

2 Feb 2022

Mapping Pu'er Tea Plantations from GF-1 Images Using Object-Oriented Image Analysis (OOIA) and Support Vector Machine (SVM)

PONE-D-21-11402R4

Dear Dr. Wang,

We’re pleased to inform you that your manuscript has been judged scientifically suitable for publication and will be formally accepted for publication once it meets all outstanding technical requirements.

Kind regards,

Daniel Capella Zanotta

Academic Editor

PLOS ONE

Additional Editor Comments (optional):

Reviewers' comments:

Reviewer's Responses to Questions

**Comments to the Author**

1. If the authors have adequately addressed your comments raised in a previous round of review and you feel that this manuscript is now acceptable for publication, you may indicate that here to bypass the “Comments to the Author” section, enter your conflict of interest statement in the “Confidential to Editor” section, and submit your "Accept" recommendation.

Reviewer #3: All comments have been addressed

2. Is the manuscript technically sound, and do the data support the conclusions?

Reviewer #3: Yes

3. Has the statistical analysis been performed appropriately and rigorously? 

Reviewer #3: Yes

4. Have the authors made all data underlying the findings in their manuscript fully available?

Reviewer #3: Yes

5. Is the manuscript presented in an intelligible fashion and written in standard English?

Reviewer #3: Yes

6. Review Comments to the Author

Reviewer #3: the authors have adequately addressed all the comments raised in a previous round of review and I feel that this manuscript is now acceptable for publication.

7. PLOS authors have the option to publish the peer review history of their article (what does this mean?). If published, this will include your full peer review and any attached files.

Reviewer #3: No

---

## [Editor Report · Acceptance letter]

7 Feb 2022

PONE-D-21-11402R4 

Mapping Pu'er tea plantations from GF-1 images using Object-Oriented Image Analysis (OOIA) and Support Vector Machine (SVM) 

Dear Dr. Wang:

I'm pleased to inform you that your manuscript has been deemed suitable for publication in PLOS ONE. Congratulations! Your manuscript is now with our production department. 

Kind regards, 

on behalf of

Dr. Daniel Capella Zanotta 

Academic Editor

PLOS ONE